# Impact of Immunopathy and Coagulopathy on Multi-Organ Failure and Mortality in a Lethal Porcine Model of Controlled and Uncontrolled Hemorrhage

**DOI:** 10.3390/ijms25052500

**Published:** 2024-02-21

**Authors:** Milomir O. Simovic, James Bynum, Bin Liu, Jurandir J. Dalle Lucca, Yansong Li

**Affiliations:** 1US Army Institute of Surgical Research, Fort Sam Houston, TX 78234, USA; msimovic@genevausa.org (M.O.S.); bynumj@uthscsa.edu (J.B.); bin.liu4.ctr@health.mil (B.L.); jurandir.dallelucca@va.gov (J.J.D.L.); 2The Geneva Foundation, Tacoma, WA 98402, USA; 3Department of Surgery, University of Texas Health Science Center at San Antonio, San Antonio, TX 78229, USA

**Keywords:** uncontrolled hemorrhagic shock, immunopathy, coagulopathy, MOF, mortality, swine, damage control resuscitation

## Abstract

Uncontrolled hemorrhage is a major preventable cause of death in patients with trauma. However, the majority of large animal models of hemorrhage have utilized controlled hemorrhage rather than uncontrolled hemorrhage to investigate the impact of immunopathy and coagulopathy on multi-organ failure (MOF) and mortality. This study evaluates these alterations in a severe porcine controlled and uncontrolled hemorrhagic shock (HS) model. Anesthetized female swine underwent controlled hemorrhage and uncontrolled hemorrhage by partial splenic resection followed with or without lactated Ringer solution (LR) or Voluven^®^ resuscitation. Swine were surveyed 6 h after completion of splenic hemorrhage or until death. Blood chemistry, physiologic variables, systemic and tissue levels of complement proteins and cytokines, coagulation parameters, organ function, and damage were recorded and assessed. HS resulted in systemic and local complement activation, cytokine release, hypocoagulopathy, metabolic acidosis, MOF, and no animal survival. Resuscitation with LR and Voluven^®^ after HS improved hemodynamic parameters (MAP and SI), metabolic acidosis, hyperkalemia, and survival but resulted in increased complement activation and worse coagulopathy. Compared with the LR group, the animals with hemorrhagic shock treated with Voluven^®^ had worse dilutional anemia, coagulopathy, renal and hepatic dysfunction, increased myocardial complement activation and renal damage, and decreased survival rate. Hemorrhagic shock triggers early immunopathy and coagulopathy and appears associated with MOF and death. This study indicates that immunopathy and coagulopathy are therapeutic targets that may be addressed with a high-impact adjunctive treatment to conventional resuscitation.

## 1. Introduction

Most battlefield casualties still die in the prehospital phase before reaching a medical treatment facility [1]. Uncontrolled hemorrhage is the leading cause of preventable deaths on the battlefield and in civilian patients [1,2]. Traumatic hemorrhage involves tissue injury, ischemia, and following reperfusion. However, these patients also suffer from perfusion and reperfusion damages that are intertwined with the complex innate immune response to trauma [3]. Current prehospital care of critically ill military and civilian trauma patients is primarily supportive of patients with traumatic hemorrhage. It does not address the destructive influence of unchecked inflammation with the potential to develop into multiple organ dysfunction syndrome (MODS) with multi-organ failure (MOF) at the end of this range [4,5,6,7]. In resource-limited settings, prehospital goal-directed resuscitation from hemorrhagic shock with low-volume hypotensive crystalloid or colloid fluid is suggested [8].

The optimal resulopathyuscitation fluid for the early treatment of severe bleeding patients remains highly debated. Roger et al. compared the rapidity of shock reversal with lactated Ringer (LR) or hydroxyethyl starch (HES) 130/0.4 at the early phase of controlled HS. They found that HES restored MAP four times faster than LR [9]. A total of 35 randomized controlled trials (RCTs) evaluating the management of traumatic HS within the first 24 h of injury did not show a correlation between transfusion requirements and mortality [9,10]. Trauma patients are at risk of hypothermia, acidosis, and coagulopathy, a fatal triad causing a vicious cycle of trauma [11]. The concept of damage control resuscitation (DCR) consists of non-surgical interventions aiming to restore homeostasis. DCR prefers blood products, but such prehospital treatment in a stern environment is logistically strenuous [https://jts.health.mil/assets/docs/damage_control/DCR_FAQs_2019-11-04_2023.pdf. accessed on 25 January 2024]; [12,13]. Damage control resuscitation combines multiple methods to prevent or reverse HS-associated events such as acidosis, hypothermia, coagulopathy, and hypoperfusion [14]. Unlike observational studies, RCTs to support the DCR approach are currently lacking [10,15]. The lack and/or inconclusiveness of clinical trials evaluating traumatic HS urges animal studies. Despite the multifactorial pathogenesis of traumatic HS, coagulopathy is the focus of scientific discourse on animal models [2,16]. One quarter to one third of patients at major trauma centers have coagulopathy on admission [15,17]. A similar proportion of coagulopathy upon admission is seen in combat patients [16]. Early TIC is portrayed as hypocoagulability that results in bleeding. In contrast, later TIC is characterized by a hypercoagulable state associated with microvascular thrombosis and MOF [18]. TIC and the effectiveness of patients’ treatment are the subjects of ongoing multifaceted debate, including reverse causality [1,15,19]. Successful resuscitation from traumatic HS is unlikely without efficient hemostasis. Intracavitary, often incompressible, and overlooked bleeding is life-threatening [20,21].

Traumatic HS, followed by fluid resuscitation, results in global ischemia and reperfusion injury, provoking a severe inflammatory response and worsening clinical outcomes [22,23]. Systemic immune system activation is fundamental to developing MOF in this context and shares many features with systemic inflammatory response syndrome (SIRS) [24]. MODS reflects a distinction between “dysfunction” and “failure” in multiple organ failure (MOF) [25], which are often interchangeably used and remain a significant cause of morbidity and mortality in trauma patients, and current therapy is based on standard supportive care. Understanding the pathophysiology of HS and resuscitation will allow for developing targeted therapeutic strategies to minimize MOF and improve patient outcomes following traumatic HS.

An emerging body of evidence indicates that crosstalk between the complement, coagulation, and fibrinolytic cascades following traumatic hemorrhage leads to microthrombosis and thromboinflammation, thereby contributing to MOF and mortality [18,26]. Central to thromboinflammation is the loss of endothelial cells’ normal antithrombotic and anti-inflammatory functions, leading to coagulopathy, complementopathy, and immunopathy. Huber-Lang et al. discovered that thrombin in a proteolytic pattern activates C5 to generate C5a without C3 involvement [27]. Amara et al. found that coagulation/fibrinolysis factors activate complement components C3 and C5, which subsequently activate and trigger the complement pathway [28]. Similarly, Gulla et al. reported that the complement is a procoagulant factor leading to thrombin activation, which generates fibrin mesh [29]. We have noted the activation of complement and coagulation cascades and their interaction, impacting outcomes in preclinical animal models of traumatic hemorrhage [30,31] and clinical trauma patients [32]. Furthermore, our recent findings have shown that the synergistic effects of the traumatic triad (complementopathy, endotheliopathy, and coagulopathy) occurred soon after trauma, contributed to poor clinical outcomes (MOF/death), and led to infection complications; therefore, the triadic intercommunication model is proposed [33]. Additionally, biomarkers such as Bb (activated factor B of the alternative pathway of complement activation), syndecan-1, and D-dimer are reliable early predictive biomarkers of clinical outcomes [33]. TIC involves many factors, including inflammatory response [16,34]. Inflammation and coagulation are reciprocally causally related processes [35]. Hemostatic resuscitation [19] may include pharmacological agents as potential adjuncts to fluid therapy to treat severe HS at or near the point of injury [36,37,38,39,40,41,42,43,44,45,46].

The purpose of this study is to characterize immune inflammatory responses, coagulation alterations, and their correlation with MOF and mortality in a lethal porcine model of controlled and uncontrolled HS with LR or HES permissive hypotensive resuscitation.

## 2. Results

### 2.1. Effect of Hemorrhage and Fluid Resuscitation on Survival

As Figure 1 presents, six-hour survival for the sham, H, H+LR, and H+Voluven^®^ groups was 9/9 (360 ± 0.0 min), 0/14 (127.6 ± 11.3 min), 11/13 (353.3 ± 6.4 min), and 1/6 (282.2 ± 27.9 min), respectively. Better survival of hemorrhaged animals treated with LR than those treated with Voluven^®^ was observed (*p* < 0.05). A clear difference in the length of survival between hemorrhaged animals and those resuscitated with fluid was observed (*p* < 0.001 and *p* < 0.01 for H vs. H+LR and H vs. H+Voluven^®^, respectively; log-rank test). There was no correlation between the length of survival and SBV at 30 min or final blood loss.

### 2.2. Baseline Characterization and Mortality

Animals were subjected to controlled bleeding (22 mL/kg b.w.) and uncontrolled splenic hemorrhage followed with or without two fluid resuscitation arms ([LR (45 mL/kg) and Voluven^®^ (15 mL/kg) at a rate of 1 mL/kg/min]. No significant differences in the hemodynamic (Table 1) and metabolic (Table 2) parameters were observed during the pre-hemorrhagic phase (baseline) among the sham, H, H+LR, and H+Voluven^®^ groups.

### 2.3. Physiological Responses to Hemorrhage

As shown in Table 1 and Table 2, the animals with HS experienced hemodynamic changes [decreased levels of pulse pressure (PP) and mean arterial pressure (MAP), and increased levels of shock index (SI)] and metabolic alterations [decreased base excess (BE) and venous oxygen saturation (SvO_2_), and increased lactate and potassium] starting at 30 min and remaining up to 120 min.

### 2.4. Effects of Fluid Resuscitation on Hemodynamic and Metabolic Parameters

As demonstrated in Table 1 and Table 2, uncontrolled shed blood volume (SBV) in swine resuscitated with LR was not significantly different from that of the H group. Resuscitation with Voluven^®^ significantly increased uncontrolled SBV at 90, 120, and 360 min compared with LR (*p* < 0.05), though there was equal uncontrolled blood loss at the pre-fluid resuscitation phase. There were no statistically significant differences between final resuscitation fluid consumption time and final urine output volume among the three groups (Appendix A). PP, MAP, and BE increased, whereas shock index (SI), blood lactate, and potassium decreased in the resuscitated animals compared with those without fluid. There was no significant difference in the hemodynamic and metabolic parameters in the LR and Voluven^®^ animals.

### 2.5. Effects of Fluid Resuscitation on Hemodilution and Coagulation Parameters

As shown in Table 2, hemoglobin (Hb) values remained constant in the sham group, decreased slightly in the H group at 90 and 120 min, and dropped significantly in other groups resuscitated with LR and Voluven^®^ at 60, 90, and 120 min. Voluven^®^ resuscitation further decreased Hb at 60, 90, and 120 min compared with the LR group. Hematocrit (Hct) values paralleled changes in Hb values (Table 2).

Thromboelastographic analysis determined whole blood coagulation function. Decreased levels of maximum amplitude (MA, Figure 2B) at 120 min and fibrinogen (Figure 2C) at 30 and 60 min, and prolonged PT (Figure 2D) at 90 min, were observed in the hemorrhaged animals. The H+Voluven^®^ group drew out a reduction in the actual strength of the clot (G, Figure 2A) and the MA (Figure 2B), reaching significance in comparison with the H+LR group, indicating a relatively hypocoagulable state (*p* < 0.05). Hemorrhaged animals resuscitated with fluids showed significantly lower fibrinogen concentrations in blood plasma (Figure 2C) and prolonged PT (Figure 2D) and PTT compared with the H group (Figure 2E). Voluven^®^ resuscitation resulted in a further increase in PT (Figure 2D) and aPTT (Figure 2E) and a further decrease in fibrinogen compared with the H+LR group (Figure 2C). Reaction time expressed as the ratio to the baseline value was not different when the H group was compared with fluid-resuscitated animals (*p* < 0.05 and *p* < 0.001 for H vs. H+LR, and H vs. H+Voluven^®^, respectively). The same was valid for K-time. There was no apparent change in the angle of the respective groups.

### 2.6. Circulating Complement Activation and Cytokine Release

Total hemolytic complement (CH50) in Voluven^®^-treated hemorrhaged swine had a significantly increased level of complement function at 60, 90, and 120 min compared with hemorrhaged non-treated animals (Figure 3A). C3a desArg, a cleavage product of systemic C3 complement component activation, was significantly increased in LR-resuscitated animals at 90 min compared with those hemorrhaged swine (Figure 3B).

Some proinflammatory and anti-inflammatory cytokines were measured in the circulation. Normalization of tumor necrosis factor-α (TNF-α) values did not result in a significant difference between the hemorrhaged only and the H+LR group. There was no apparent difference in the TNF-α blood concentrations between the H+Fluid groups nor between the H+LR and H+Voluven^®^ groups (Figure 3C). Systemic IL-6 levels were also increased in Voluven^®^-treated animals at 120 min compared with LR-treated swine (Figure 3D). As a result of the significant differences among the individual IL-8 concentrations at the baseline, plasma levels of IL-8 are shown as a percent of the baseline to permit a direct comparison of relative changes after HS. When IL-8 concentration was normalized to the baseline values, a significant difference in its levels between respective groups could not be observed (Figure 3E). Anti-inflammatory cytokines such as IL-10 were undetected in the blood. 

### 2.7. End Organ Function

Creatinine blood levels were significantly elevated in Voluven^®^-treated hemorrhage swine at 60, 90, and 120 min after the onset of injury compared with the H+LR group (Figure 4A). Aspartate aminotransferase, a marker of hepatocellular damage, was significantly higher in injured and Voluven^®^-treated swine than in the H+LR group (Figure 4B) at 120 min. There was no statistically significant difference in muscle myocardium isoenzyme B (MMB) levels between hemorrhaged animals and those injured and fluid-resuscitated (Figure 4C). Independently of experimental condition, creatine kinase (CK) levels did not show significant changes (Figure 4D).

### 2.8. Myocardial Inflammatory Responses to Hemorrhagic Shock and Fluid Resuscitation

As illustrated in Figure 5, there was increased deposition of C4d, C3, C5, C5b-9, and IL-6 in porcine hearts after exposure to hemorrhagic shock. Hemorrhaged animals with LR resuscitation were not affected by these protein depositions compared with the H group. Voluven^®^-resuscitated animals had significantly higher levels of C5b-9 in their hearts than the groups with H and H+LR (*p* < 0.05).

### 2.9. Pulmonary and Intestinal Inflammatory Responses to Hemorrhagic Shock and Fluid Resuscitation

As demonstrated in Figure 6, enhanced expression of C3 and C5b-9 in the jejunum tissue was observed in hemorrhaged animals with or without fluid resuscitation. There was no statistically significant difference among the three groups (H, H+LR, and H+Voluven^®^). Hemorrhage significantly increased intestinal deposition of C3 and IL-6. Fluid resuscitation had little impact on the intestinal tissue levels of C3 and IL-6 except for LR resuscitation on C3 expression.

### 2.10. Effect of Hemorrhage and Fluid Resuscitation on Organ Histopathological Alterations

The exposure to the hemorrhagic shock w or w/o LR resuscitation resulted in pathological changes typical of mild myocarditis characterized by monocyte and neutrophil infiltration (white rectangle), whereas Voluven^®^ resuscitation tended to cause more inflammatory cell infiltration but did not reach a statistically significant difference (Figure 7A,B). H, H+LR, and H+Voluven^®^ induced moderate–severe pulmonary (septal thickening, inflammatory cell infiltration, alveolar hemorrhage, and edema) and intestinal (denuded villi with lamina propria exudate) damages. Still, there were no statistically significant differences among the three groups (Figure 7C–F). H and H+LR led to renal damage (proximal tubular epithelial cell hydropic degeneration, border brush loss, and interstitial inflammatory cell infiltration) (Figure 7G,H). Animals resuscitated with LR and Voluven^®^ had improved and worsened renal damage, respectively (Figure 7G,H). Hepatic injury (moderate hepatic cell apoptosis and degeneration, vascular congestion and inflammatory cell infiltration) was observed in animals subjected to hemorrhage, while fluid resuscitation with LR or Voluven^®^ did not attenuate liver injury (Figure 7I,J).

### 2.11. Correlation between Early Immunopathy and Coagulopathy and Organ Injury

There was an inverse correlation between blood levels of fibrinogen at 60 min after hemorrhage and liver tissue injury score (Figure 8A, Spearman *r* = −0.59, n = 12, *p* = 0.0463) and CH50 at 120 min after hemorrhage and kidney tissue injury score (Figure 8B, Spearman *r* = −0.86, n = 7, *p* = 0.0238), and a positive correlation between CH50 at 30 min after hemorrhage and lung tissue injury score (Figure 8C, Spearman *r* = 0.60, n = 12, *p* = 0.0423).

## 3. Discussion

Hemorrhage after severe trauma remains the leading cause of potentially preventable death in young individuals (≤45 years) in traumatically injured civilian (~40%) and military populations (~50%) [47,48,49]. In combat, 87% of battlefield deaths after HS occur before reaching a medical facility. Outcomes of patients with HS and extremity bleeding have improved with the widespread use of tourniquets and early balanced transfusion therapy. However, point-of-injury and prehospital care of injured patients with uncontrolled truncal hemorrhage is a challenge and has had the same mortality over the last two decades despite recent advances in treatment protocols, including tourniquets, permissive hypotension, point-of-care ultrasonography, tranexamic acid, massive transfusions, and all being performed within the “golden hour” [50].

In this study, we investigated the effect of fluid resuscitation with lactated Ringer’s solution or Voluven^®^ on porcine combined controlled and uncontrolled bleeding. Swine were surveyed 6 h after completion of splenic hemorrhage or until death. Hemorrhagic shock resulted in systemic and local complement activation, cytokine release, coagulopathy, metabolic acidosis, hyperkalemia, MOF, and no animal survival. Compared with the LR group, the animals with hemorrhagic shock treated with Voluven^®^ had worse dilutional anemia, coagulopathy, renal and hepatic dysfunction, increased myocardial complement activation and renal damage, and decreased survival rate.

We correlated tissue injuries (MOF markers) in the lung, heart, gut, liver, and kidney tissue with IL-6, thrombotic parameters such as PT, PTT, and fibrinogen blood levels, and CH50 in the hemorrhage group alone. The positive correlation between CH50 assayed 30 min after hemorrhage completion and lung tissue injury is puzzling. The tissue was harvested at the latest 209 min after hemorrhage, the most extended survival of an animal in the hemorrhagic shock group. The complement activation occurred earlier, priming tissues before blood sampling at 30 min when the complement hemolytic capability (CH50) had recovered considerably. In a porcine hemorrhagic shock model, CH50 dropped during the shock phase, showing excessive complement activation before reperfusion occurred. Resuscitation with plasma expanders induced an additional 20% of complement consumption. Whole blood transfusion subsequently increased CH50 values [51].

Lactate blood levels in fluid-resuscitated swine were significantly lower 90 and 120 min after hemorrhage compared with lactate plasma concentrations in non-resuscitated animals. It is assumed that lactatemia reduction is due to better tissue perfusion in fluid-resuscitated animals. However, hyperlactatemia can occur even during satisfactory tissue perfusion and oxygenation [52]. Apart from other parameters, lactate blood concentration is a marker for bleeding. Investigating the chronological order of occurrence of bleeding markers, Treml et al. found lactate increases immediately after shedding blood in a porcine exsanguination model [53].

An experimental model developed and run through experimentation is crucial in animal studies. This is especially true for hemorrhagic shock models, as there is no reliable reproduction of the clinical circumstances [54,55,56]. Our study does not include splenectomy, and here, we debate using splenectomy in modeling hemorrhagic shock. Unlike that of a human, a pig’s spleen sequesters up to 20–25% of the animal’s blood volume and can auto-transfuse blood rapidly under severe hemorrhagic conditions. Histological splenic examination indicated successful auto-transfusion, at least in the recently dead pig. The spleens from the pigs that survived the observation period (24 h) had erythrocytes in the red pulp, but this fact cannot reflect the organ dynamics at the time close to injury [57]. Whether splenectomy is an essential procedure in porcine hemorrhage studies is an enduring issue [58,59], and caution needs to be exercised when bleeding is higher than 30% [60]. Vnuk et al., using a volume-controlled hemorrhagic porcine model and specified anesthetics (azaperone, thiopental, and isoflurane), showed that sham-operated animals were hemodynamically more stable than splenectomized animals and those with an auto-transplanted spleen [61]. We recognize potential differences among pigs with intact spleens that can transfuse irregular volumes of blood into circulation. The variation in the capacity of the spleen to permeate different volumes of blood is a categorical property of the animal’s body, independent of its environment. Therefore, this variability should be counted on despite the possible requirement of a relatively more significant number of animals for testing. Pottecher et al. underlined that the pre-hemorrhagic porcine splenectomy model only reproduces the situation when hemorrhagic shock follows elective surgical splenectomy [62].

Hemorrhaged patients after severe trauma are particularly susceptible to the early development of coagulopathy, immunopathy [complementopathy, systemic inflammatory response syndrome (SIRS), immunoparesis], endotheliopathy, and metabopathy (tissue hypoperfusion-induced energy metabolic dysfunction and iron metabolic alterations) on admission to the hospital [26,32,37,42,45,46,63,64,65,66,67,68]. Moreover, trauma-induced exposure of tissue factor to flowing blood induces the activation of coagulation, which may trigger consumptive coagulopathy [18]. Uncontrolled bleeding after trauma initiates coagulopathy via loss of coagulation factors, red blood cells, and platelets, coagulation cascade activation, immunopathy, endotheliopathy, and metabopathy through global ischemia. Moreover, therapeutic approaches (e.g., fluid resuscitation, blood transfusion, surgical procedures, extracorporeal life support devices) can further worsen these multi-opathies and perpetuate bleeding [69].

Severe HS is a system failure, including coagulation, immunity, vital organs, vasculature, endothelium, mitochondria, and physiological barriers [70]. Post-HS systemic uncoupling of coagulation, immunity, vital organ vasculature, endothelium, mitochondria, and physiological barriers is critical in morbidity and mortality. Maintaining oxygen delivery to limit tissue hypoxia, inflammation, and organ function is essential for uncontrolled HS. Damage control resuscitation (DCR) recommends using more blood products and fewer clear fluids (crystalloid and colloid solutions) for initial resuscitation in treating HS. However, prehospital resuscitation in austere environments largely relies on crystalloid and colloid intravascular expansion, as blood products are logistically arduous. Ordinary fluid resuscitation in such resource-limited conditions is the first therapeutic intervention to replace blood lost and preserve tissue perfusion until definite surgical control of bleeding can be achieved. Plain fluid resuscitation can worsen clinical outcomes by increasing blood loss by elevating blood pressure, dislodging blood clots, diluting coagulation factors and platelets, and causing inflammation and coagulopathy [71].

Our previous studies have shown controlled hemorrhagic shock in rats [45,46] and pigs [39], with typical fluid resuscitation triggering tissue hypoperfusion, metabolic acidosis, systemic and local inflammation, and physiological barrier dysfunction, leading to multiple organ damage and death. Even though DCR with LR or Voluven^®^ following HS improves some measures of hemodynamics, metabolism, hyperkalemia, and survival in this current study, these restricting fluid therapies result in further complement activation and worse coagulopathy. The most important advantages of using colloids are the rapid achievement of hemodynamic goals because of their slow diffusion into the extravascular space and logistical advantages due to reducing the weight and volume of resuscitation. Some colloids have been attributed as causing additional harmful effects on hemostasis, with altered fibrin polymerization and decreased platelet adhesive and aggregating properties [9]. Clinical data from a series of large and randomized controlled trials in critically ill patients failed to show an outcome advantage associated with colloidal fluid resuscitation and indicate that hydroxyethyl starch solutions may be related to significant adverse effects (e.g., acute kidney injury, need for renal replacement therapy, coagulopathies, and pathological tissue uptake) [72]. 

We previously demonstrated that Hextend^®^ colloid infusion significantly contributes to dilutional anemia, tissue inflammation, complement activation, multiple organ damage, and mortality, although it boosts hemodynamics after controlled HS in swine [39]. In this current study, Voluven^®^ resuscitation led to more uncontrolled bleeding, worse dilutional anemia, systemic inflammation, coagulopathy, renal and hepatic dysfunction, increased myocardial complement activation, and renal damage. It decreased the survival rate compared with balanced LR crystalloid infusion. As such, developing optimal DCR by targeting immunopathy and/or coagulopathy may be a promising adjunctive strategy for prehospital settings.

In recent years, immunological damage control resuscitation has shown promise as a pharmacological agent for use in trauma [42,73,74,75,76,77], traumatic HS [78,79,80], and sepsis [81,82]. Our recent works have shown (1) the beneficial effects of complement inhibition as both a stand-alone and adjunctive treatment with HS [39,43,83,84] and (2) the evident effectiveness of immunological damage control resuscitation even in complex animal models of combined HS and polytrauma [37,45,46]. Prehospital treatment with tranexamic acid (TXA) has been shown to reduce mortality in a large international trauma study [85]; it has been recommended in some prehospital systems and included in the WHO list of essential medicines for the treatment of trauma [World Health Organization. Summary of the Report of the 18th Meeting of the WHO Expert Committee on the Selection and Use of Essential Medicines. https://www.who.int/selection_medicines/committees/TRS_web_summary.pdf (2021). accessed on 26 January 2014]. Tranexamic acid-mediated plasmin inhibition modulated the immune system and diminished surgery-set-off immunosuppression following cardiac surgery in patients [86]. Relke et al. provided an illustrated review of clinical TXA evidence and controversies [87]. Okholm et al. reported in a review of human studies that the anti-inflammatory effect of TXA was consistently found only among orthopedic patients [88]. The prehospital use of TXA is still controversial because there was no significant difference in the mortality with TXA versus a placebo in multicenter RCTs [89,90]. However, patients with severe shock who received early prehospital administration of TXA had a significant reduction in 30-day mortality [91], suggesting the severe traumatic HS patient group would benefit the most from TXA. Altogether, in the future, hemostatic resuscitative crystalloid and colloid fluids in combination with pharmacological agents to deter coagulopathy and immunopathy may resolve some of these problems and show more outcome benefits.

Our study has limitations, such as pairing controlled with uncontrolled bleeding. This approach may deserve further consideration to benefit from the advantages of the two methods. Enrolling only female pigs is another limitation. However, gender differences in trauma and shock are still debated [92]. Swine are hypercoagulable compared with humans [93], and there are limitations to animal models of trauma-induced coagulopathy [94]. Using a homeothermic blanket in an experimental setting, as we used, does not reflect the actual clinical situation, and we do not know how much laparotomy could compensate for this setback. In reality, hemorrhagic shock is followed by hypothermia that primes platelets [95]. Laparotomy can be required in the treatment of postpartum hemorrhage [96], but laparotomy in the context of our study may require further evaluation.

## 4. Conclusions

Controlled and uncontrolled hemorrhage-triggered immunopathy and coagulopathy appear associated with increased MOF and mortality. Further characterization of immunopathy and coagulopathy in a larger sample size with severe hemorrhagic shock is warranted. This study suggests that (1) LR, a crystalloid solution, is a better agent than Voluven^®^, a colloid solution; (2) the development of adjunctive strategies by targeting early immunopathy and coagulopathy may be a promising therapy for patients with trauma and hemorrhagic shock in point-of-injury care and on-scene care.

## 5. Materials and Methods

### 5.1. Animal Study

This research complied with the Animal Welfare Act and the implemented Animal Welfare regulations, and the study was conducted in compliance with the Animal Welfare Act, the implemented Animal Welfare regulations, and the principles of the Guide for the Care and Use of Laboratory Animals, National Research Council. The facility’s Institutional Animal Care and Use Committee approved all research conducted in this study (approved code: A-10-017; approved date: 23 September 2012). The facility where this research was conducted is fully accredited by AAALAC International.

#### 5.1.1. General Procedures

Yorkshire-cross female pigs [specific pathogen-free weighing 38.9 ± 2.9 kg (mean ± SD)] were obtained from Midwest Swine Research (Gibbon, MN, USA) and held in an AAALAC International-accredited facility, and their acclimatization lasted a minimum of 5 days. After this period, baseline hematologic and biochemical screening was performed.

#### 5.1.2. Surgical Preparation

The pigs were fasted for 12 to 18 h with free access to water before surgery. Pre-surgical procedures and instrumentation were applied as in our previous report [97]. In brief, preterminal anesthesia medication for secretion control (glycopyrrolate, Robinul, 0.01 mg/kg, Baxter Healthcare, Deerfield, IL, USA) and sedation (tiletamine-zolazepam, Telazol^®^, 8 mg/kg, Wyeth, Fort Dodge, IA, USA) were administered intramuscularly. After intubation and anesthesia, the animals were supine on a standard operating table. Surgical instrumentation was conducted under anesthesia with 1.5–2.5% isoflurane in 30% oxygen in the air using an automatic ventilator and monitor (Fabius GS gas anesthesia system and Infinity Delta XL monitoring system, Draeger Medical, Telford, PA, USA). An end-tidal pCO_2_ of approximately 40 mm Hg was maintained. Core temperature was supported throughout by a homeothermic blanket and forced-air warming system. Urine was collected transurethrally using all silicone Foley catheters (10 Fr., 3-mL balloon, Sherwood Medical, St. Louis, MO, USA).

Vascular catheters were inserted via cut-downs. A micromanometer (SPC 330A; Millar Instruments, Inc., Houston, TX, USA) was inserted non-occlusively into the left internal carotid artery for blood pressure and heart rate monitoring. A Swan-Ganz catheter was passed via the left jugular vein and positioned with its tip in the pulmonary artery, as confirmed by insertion tracings for continuous measurement of cardiac output and mixed venous oxygen saturation (Vigilance II Monitor, Edwards Lifesciences LLC, Irvine, CA, USA). Occlusive catheters (8 Fr. Sideport/percutaneous catheter introducer, Argon Medical Devices, Athens, TX, USA) were inserted into the left femoral artery and left femoral vein for hemorrhage and fluid infusion, respectively. The arterial and venous blood samples were collected via these catheters. Another non-occlusive micromanometer was also put into the left femoral artery through the catheter introducer. These occlusive catheters were maintained patent by a slow continuous infusion of saline through intraflow adapters (3 mL/h, intraflow continuous flush devices, Abbott Laboratories, Abbott Park, IL, USA), which were connected to bags of normal saline pressurized to 300 mmHg. Access to the spleen was provided via a laparotomy, and a plastic sheet was placed between the spleen and the intestines. After the controlled hemorrhage, laparotomy was performed distally of the umbilicus with a suprapubic single incision. With a skin marker, a line was made down the entire length of the spleen 1 cm lateral to the midline to avoid injuries to large arteries and veins in the spleen. Most of the splenic bleeding was complete by 15 min when the laparotomy was closed. These and other steps in inflicting uncontrolled hemorrhage were described earlier [98].

#### 5.1.3. Experimental Design

Once the instrumentation was completed and the vital parameters stabilized, the animals were subjected to somewhat modified versions of the procedures earlier reported [98]. A 10 min baseline began, and hemodynamic measurements were collected from the analog and RS-232 signals on a data acquisition instrumentation rack (Dynamic Research Evaluation Workstation–DREW, US Army Institute of Surgical Research, San Antonio, TX, USA).

Figure 9 shows the conduct of the experiments. The animals were enrolled in one of four experimental groups: (1) control, sham-operated (cannulated but not hemorrhaged/injured), n = 9; (2) H, hemorrhage + splenic injury, n = 14; (3) H + LR, hemorrhage + splenic injury + lactated Ringer’s solution (LR), n = 13; and (4) H + Voluven^®^, hemorrhage + splenic injury + Voluven^®^, n = 6.

A controlled hemorrhage (22 mL of blood/kg b.w. 100 mL/min) was performed using our custom error-sensing negative feedback computerized pump program [97]. After controlled hemorrhage, the animals underwent partial splenic resection [98]. The uncontrolled hemorrhage volume from the injured spleen was measured continuously by suctioning shed blood into canisters (Vac-Rite disposable suction system, Baxter Healthcare, Deerfield, IL, USA) that had been placed on a balance (SR16000 Mettler Balance, Mettler-Toledo, Columbus, OH, USA). The time the splenic injury was completed was marked as the zero-time point.

Fluid resuscitation was performed 30 min after the splenic injury was completed, marked as a zero-time point (Figure 8). Resuscitation started with intravenous infusion with crystalloid (lactated Ringer’s solution; LR) at 45 mL/kg body weight or colloid solution (Voluven^®^, 6% hydroxyethyl starch 130/0.4 in 0.9% sodium chloride injection; Fresenius Kabi Norge AS, Halden, Norway) at 15 mL/kg body weight at a rate of 1 mL/kg/min using our servo-controlled computerized pump. Animals were observed for 6 h after the injury or until death.

#### 5.1.4. Biosampling

As seen in Figure 9, an arterial (30 mL) blood sample was obtained before controlled hemorrhage (−30 min), after shock, and immediately before resuscitation (30 min), continued at 30 min intervals within the first 120 min, and followed by hourly sampling. Baseline screening was not performed before surgical instrumentation. A “baseline” blood sampling was made after blood vessel catheterization but before controlled hemorrhage and partial splenic resection. The acid–base balance in the blood was assayed using i-STAT cartridges (Abbott Laboratories, Abbott Park, IL, USA). At the end of the observation or death (when end-tidal pCO_2_ ≤ 10 mmHg or at flat-line ECG), each animal was euthanized with sodium pentobarbital (90 mg/kg IV, 10 mL Fatal Plus, Vortech Pharmaceuticals, LTD, Dearborn, MI, USA) under surgical anesthesia.

Tissue harvest: Tissue samples, including the left lung, jejunum, left central liver lobe, left heart ventricle, and left kidney, were removed and fixed with 10% formalin or 4% paraformaldehyde for histological and immunohistochemical analysis.

### 5.2. Assays

#### 5.2.1. Reagents and Antibodies

CG4+ and CG8+ cartridges were purchased from Abbott (Princeton, NJ, USA). Chicken anti-C3/C3a, mouse anti-C4d, mouse anti-C5b-9, and mouse anti-endothelial cell antibodies were obtained from Abcam Inc. (Cambridge, MA, USA). Rabbit anti-C5 antibody was from Abbiotec, LLC (San Diego, CA, USA). Mouse anti-IL-6 was purchased from R&D Systems (Minneapolis, MN, USA). Mouse anti-porcine C3a, biotinylated anti-C3a, and porcine C3a standard were from Georg-August-Universität Göttingen Stiftung Öffentlichen Rechts (Göttingen, Germany). Goat anti-chicken Alexa Fluor 594, goat anti-mouse Alexa Fluor 488, goat anti-mouse Alexa Fluor 594 IgG (H+L) conjugated secondary antibodies, and ProLong Gold antifade reagent were from Invitrogen (Carlsbad, CA, USA).

#### 5.2.2. Histological Examination

Formalin-fixed tissues were embedded in paraffin, sectioned, and stained with hematoxylin–eosin as we described previously [30,37,39,83,84]; histological images (3 sections/tissue and 5 filed/section) were recorded under a light microscope (AX80; Olympus, Center Valley, PA, USA) by a pathologist blinded to the treatment group. Histological injury scores were graded according to the following:

For lung injury, four parameters (alveolar fibrin edema, alveolar hemorrhage, septal thickening, and intra-alveolar inflammatory cells) were scored on each H&E-stained slide for (1) the severity (0: absent; 1, 2, and 3 for more severe changes) and (2) the extent of injury (0: absent; 1: <25%; 2: 25–50%; 3: >50%) for the lung tissue.

Each slide’s mucosal damage of the small intestine was graded on a six-tiered scale. A score of 0 was assigned to a normal villus; villi with tip distortion were scored as 1; villi lacking goblet cells and containing Guggenheims’ spaces were scored as 2; villi with patch disruption of the epithelial cells were scored as 3; villi with exposed but intact lamina propria and epithelial cell sloughing were assigned a score of 4; villi in which the lamina propria was exuding were scored as 5; villi displaying hemorrhage or denuded were scored as 6.

Myocardial injury was assessed by using four parameters (edema, degeneration, inflammatory cell infiltration, congestion) that were scored on each H&E-stained slide for (1) the severity (0: absent; 1, 2, and 3 for more severe changes) and (2) the extent of injury (0: absent; 1: <25%; 2: 25–50%; 3: >50%).

For the evaluation of liver injury, four parameters were used. Vascular congestion/thrombosis is defined as engorgement of portal venules, sinusoids, or terminal hepatic venules with erythrocytes, platelets, and/or fibrin material (score 0 for no change, scores 1, 2, 3 for more extended and severe changes). Hepatocyte death is assessed by loss of nuclear detail and well-defined cellular borders (score 0 for no change, scores 1, 2, 3 for more extended and severe changes). Degeneration is determined by cytoplasmic hydropic change, cytoplasmic vacuolization, and sinusoid derangement (score 0 for no change, scores 1, 2, 3 for more extended and severe changes). Inflammation is evaluated by inflammatory cell infiltration such as by macrophages, histocytes, and polymorphonuclear leukocytes (PMN) (score 0 for no change, scores 1, 2, 3 for more extended and severe changes). The extent of liver damage (0: absent; 1: <25%; 2: 25–50%; 3: >50%) was also evaluated.

Kidney injury was evaluated by using the following scores: 0 = normal histology; 1 = slight alteration (loss of brush border, mild hydropic degeneration, mild congestion); 2 = mild (intensive hydropic degeneration, mild vacuolization, interstitial edema); 3 = moderate (nuclear condensation, intensive vacuolization, modulated interstitial edema); 4 = severe (necrotic/apoptotic cells, denudation/rupture of basement membrane); 5 = necrosis (total necrosis of the tubule). They were scored on each H&E-stained slide for (1) the severity (0: absent; 1, 2, and 3 for more severe changes) and (2) the extent of injury (0: absent; 1: <25%; 2: 25–50%; 3: >50%).

#### 5.2.3. Immunohistochemical Staining

As described previously [37,39,83,84], paraformaldehyde-fixed lung and small intestine biopsies were snap-frozen at −70 °C; sections were cut at a 5 µm thickness with a cryostat and fixed in cold methanol for 20 min. The fixed sections were permeabilized with 0.2% Triton X-100 in PBS for 10 min and then blocked with 2% BSA in PBS for 30 min at room temperature. The sections were incubated with the primary antibodies (anti-C5b-9, C3a, and C5a) overnight at 4 °C, washed, and then incubated with the appropriate secondary antibodies labeled with Alexa Fluor 488 and 594 for 1 h at room temperature. After washing, the sections were mounted with ProLong Gold antifade solution containing 4′, 6′-diamidino-2-phenylindole (DAPI) and visualized under a confocal laser scanning microscope (Radiance 2100; Bio-Rad, Hercules, NJ, USA) at ×400 magnification. Negative controls were conducted by substituting the primary antibodies with corresponding immunoglobulin isotypes. Captured digital images were processed with Image J software (Version v2.0, NIH, Bethesda, MD, USA). As previously described, a modified method quantified immunofluorescent intensity [99]. Four to six images from each animal were opened using Adobe Photoshop software (Version: 6.0, San Jose, CA, USA) and adjusted until only the fluorescent deposits and no background tissue were visible. The image was changed to black and white pixels using Image J software, with black representing the target proteins’ deposits and white representing non-stained areas of the image. The image was then changed to red and white using the Adjust Threshold command, with fluorescent deposits being red. The image was analyzed to show the total red area in pixels squared. Values for the entire area for all animals in each group were averaged to give the average area of the fluorescent deposit.

#### 5.2.4. Cytokine Assays

Serum TNF-α, IL (Interleukin)-6, and IL-8 levels were assessed using a quantitative sandwich enzyme-linked immunosorbent assay (ELISA; R&D Systems, Minneapolis, MN, USA) according to the manufacturer’s instructions.

#### 5.2.5. Quantitative Assessment of End Tissue Function

Creatinine, aspartate aminotransferase (AST), muscle myocardium isoenzyme B (MMB), and creatinine kinase (CK) were measured from blood samples using the Dimension Xpand Plus Integrated Chemistry System (Siemens, Holliston, MA, USA).

#### 5.2.6. Measurement of Coagulation Parameters

Samples for coagulation assays were collected in citrated tubes. Prothrombin time (PT), activated partial thromboplastin time (PTT), and fibrinogen concentrations were measured in platelet-poor plasma using the BCSTM XP system (Siemens, Deerfield, IL, USA). Whole blood coagulation function was determined with Thrombelastography™ (TEG) (Haemoscope Corporation, Niles, IL, USA).

#### 5.2.7. Analysis of Complement Functional Activity

Serum complement activity was determined based on hemolytic activity [30,37]. Briefly, antibody-sensitized *Gallus gallus domesticus* red blood cells (Colorado Serum Company, Denver, CO, USA) were incubated for one hour at 37 °C with serial dilutions of serum samples in gelatin–Veronal buffer (pH 7.3). After centrifugation, the absorbance of the supernatant was determined at 405 nm, and the serum concentration inducing 50% of complement hemolytic activity was defined as a CH50 value.

#### 5.2.8. Analysis of Plasma C3a

As described previously [30,37], 96-well microplates (R&D Systems) were coated with anti-porcine C3a mAb (Georg-August-Universität Göttingen Stiftung Öffentlichen Rechts, Germany) in coating buffer (Na_2_CO_3_/NaHCO_3_ coating buffer, pH >10) overnight at 4 °C. After blocking with 1% gelatin in coating buffer and washing three times with wash buffer (1× PBS containing 0.05% Tween 20), 100 µL of standards (porcine C3a) or porcine plasma samples was added into each well and incubated for 2 h at room temperature. After incubation, the wells were washed three times with wash buffer. Then, 100 µL of biotinylated anti-porcine-C3a IgG (K5/4, Georg-August-Universität Göttingen Stiftung Öffentlichen Rechts, Germany) was added and incubated for one hour at room temperature. Wells were washed and incubated with streptavidin-HRP for one hour at room temperature, followed by incubation with 100 μL of substrate (R&D Systems) for 20 min at room temperature. After incubation, 50 μL of stop solution (R&D Systems) was added to each well and read at 450 nm with a plate reader.

#### 5.2.9. Serum Protein Assay

Levels of total protein in plasma were measured using a bicinchoninic acid protein assay kit (cat#23225, Pierce, Rockford, IL, USA) according to the manufacturer’s instructions.

### 5.3. Statistical Analysis

Data were analyzed using GraphPad Prism version 10 (GraphPad Software, La Jolla, CA, USA) and are expressed as mean ± SEM. The log-rank (Mantel–Cox) test was used to analyze survival. Hemodynamic and metabolic parameters were analyzed using two-way ANOVA with Bonferroni posttests, while thromboelastographic and histological data were reviewed with the Kruskal–Wallis test; *p* < 0.05 was considered significant. Mixed-effect analysis evaluated PT, PTT, and fibrinogen levels in the hemorrhage group.

## Figures and Tables

**Figure 1 ijms-25-02500-f001:**
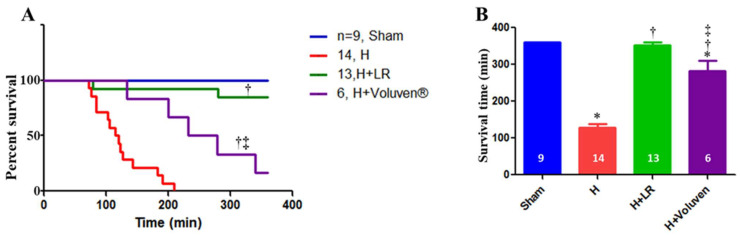
Survival in a porcine model of hemorrhagic shock and fluid resuscitation. Percent survival (**A**) and survival time are shown. * *p* < 0.05 vs. sham, † *p* < 0.05 vs. H, and ‡ vs. H+LR, *p <* 0.05 using a log-rank test (**A**) and one-way ANOVA followed by Bonferroni posttests (**B**), respectively.

**Figure 2 ijms-25-02500-f002:**
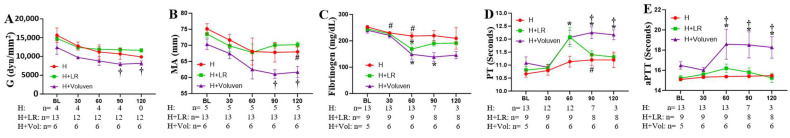
Coagulation disturbances after hemorrhagic and fluid resuscitation. Whole blood shear elastic modulus (G) parameter (**A**) and maximum amplitude (MA, (**B**)) were measured by thromboelastography. Fibrinogen concentrations (**C**), plasma prothrombin time (PT, (**D**)), activated partial thromboplastin time (aPTT, (**E**)), and were assessed using BCSTM XP system. * H+Voliuven/H+LR vs. H and † H+Voluven vs. H+LR, *p* < 0.05 using two-way ANOVA followed by Bonferroni posttests. # vs. BL, *p* < 0.05, using unpaired *t*-test (two-tailed). Data are presented as mean ± SEM. BL, baseline; PT, extrinsic pathway; aPTT, contact pathway.

**Figure 3 ijms-25-02500-f003:**
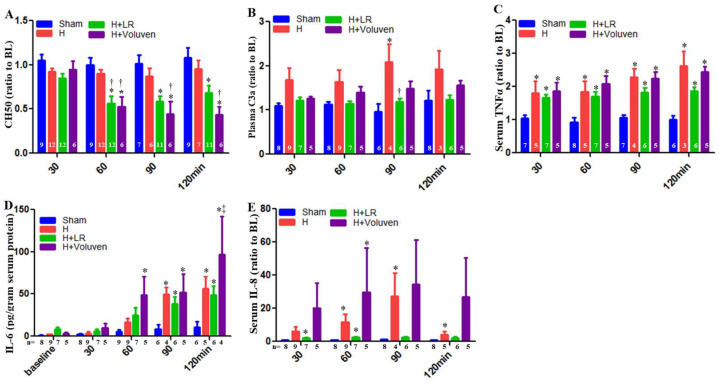
Systemic inflammatory immune responses after hemorrhagic shock and fluid resuscitation. Serum hemolytic terminal complement activation was measured with CH50 assay (**A**), and blood levels of C3a (**B**), TNαF (**C**), IL-6 (**D**), and IL-8 (**E**) were assessed with ELISA. * vs. sham, † vs. H, and ‡ vs. H+LR, *p* < 0.05 using two-way ANOVA followed by Bonferroni posttests. Data are presented as mean ± SEM.

**Figure 4 ijms-25-02500-f004:**
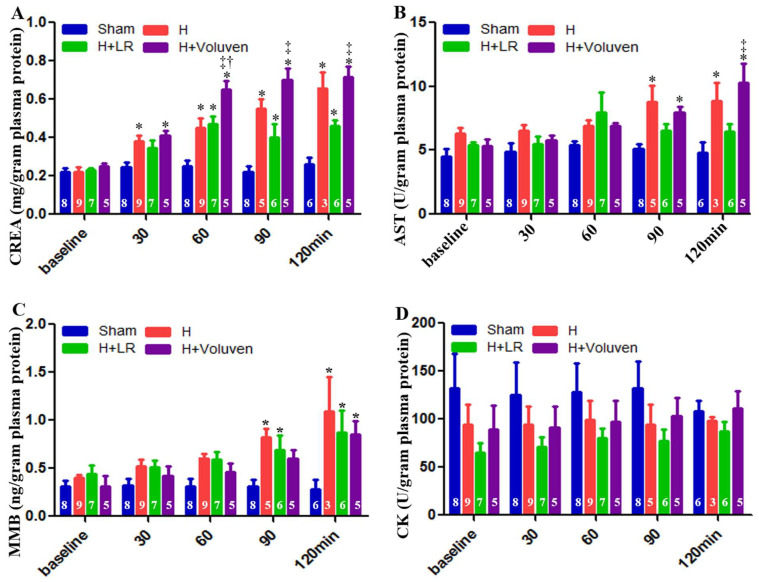
Multiple organ dysfunctions post-hemorrhagic shock and post-fluid resuscitation. Blood creatinine (**A**), aspartate aminotransferase (AST, (**B**)), muscle myocardium isoenzyme B (MMB, (**C**)), and creatine kinase (CK, (**D**)) were determined with Siemens Dimension Xpand Plus Chemistry 6 Analyzer. * vs. sham, † vs. H, ‡ vs. H+LR, *p* < 0.05 using two-way ANOVA followed by Bonferroni posttests. Data are presented as mean ± SEM.

**Figure 5 ijms-25-02500-f005:**
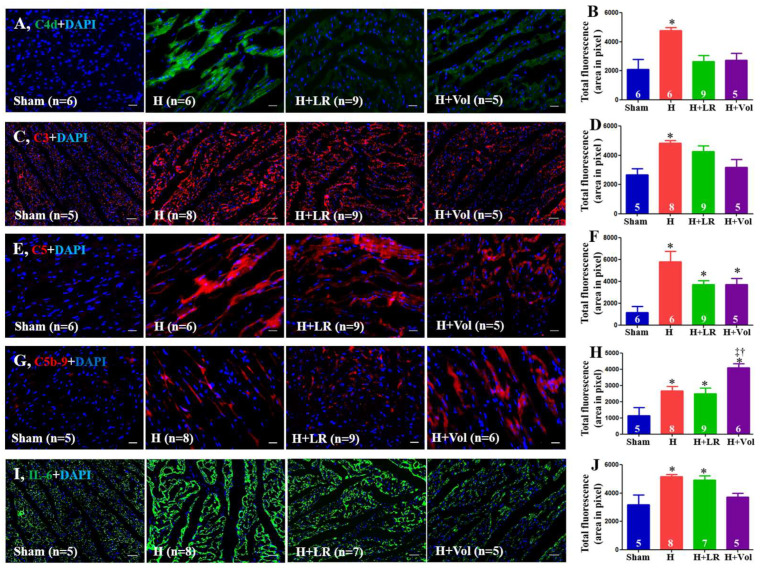
Myocardial inflammatory responses after hemorrhagic shock and fluid resuscitation. Immunostaining and semiquantitative fluorescent intensity of C4d (**A**,**B**), C3 (**C**,**D**), C5 (**E**,**F**), C5b-9 (**G**,**H**), and IL-6 (**I**,**J**) in heart tissues were evaluated by immunohistochemistry. Scale bars = 50 μm. * vs. sham, † vs. H, and ‡ vs. H+LR, *p <* 0.05 using one-way ANOVA followed by Bonferroni posttests. Data are presented as mean ± SEM. Vol, Voluven. DAPI = 4′, 6′-diamidino-2-phenylindole.

**Figure 6 ijms-25-02500-f006:**
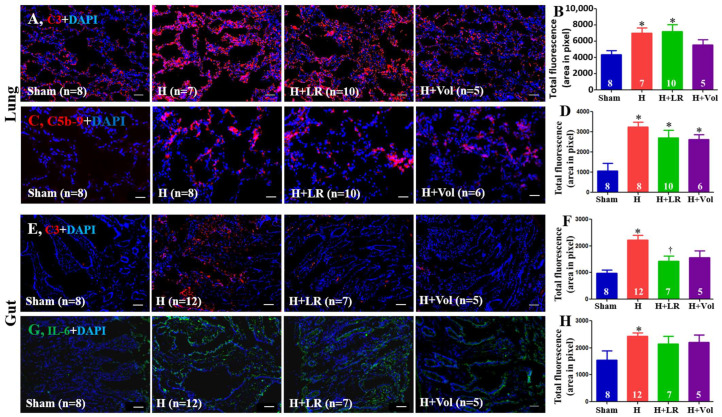
Pulmonary/intestinal inflammatory responses after hemorrhagic shock and fluid resuscitation. Immunostaining and semiquantitative fluorescent intensities of C3 (**A**,**B**) and C5b-9 (**C**,**D**) in lungs and C3 (**E**,**F**) and IL-6 (**G**,**H**) in jejunum were evaluated by immunohistochemistry. Scale bars = 50 μm. * vs. sham, † vs. H, *p <* 0.05 using one-way ANOVA followed by Bonferroni posttests. Data are presented as mean ± SEM.

**Figure 7 ijms-25-02500-f007:**
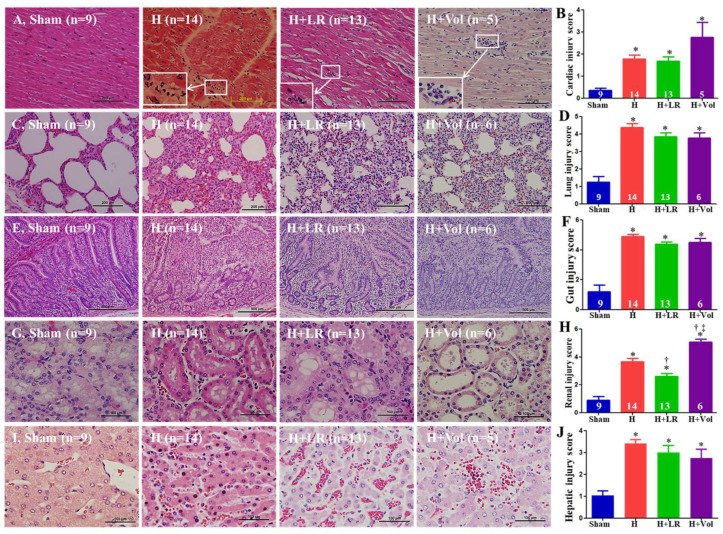
Histopathological changes in end organs after hemorrhagic shock and fluid resuscitation. Histopathological (H&E stain) photos and semiquantitative evaluations of the heart ((**A**,**B**), magnification = ×200, scale bars = 200 µm), lung ((**C**,**D**), magnification = ×200, scale bars = 200 µm), jejunum ((**E**,**F**), magnification = ×100, scale bars = 500 µm), kidney ((**G**,**H**), magnification = ×400, scale bars = 100 µm), and liver ((**I**,**J**), magnification = ×400, scale bars = 100 µm). Myocarditis is marked with white arrows (**A**), and the insert in panel (**A**) magnifies the region of the indicated box to show the mononuclear cells and polymorphic nuclear cells in the corresponding inflammatory infiltration foci. Data are presented as mean ± SEM. * vs. sham, † vs. H, and ‡ vs. H+LR, *p <* 0.05, using one-way ANOVA followed by Bonferroni posttests.

**Figure 8 ijms-25-02500-f008:**
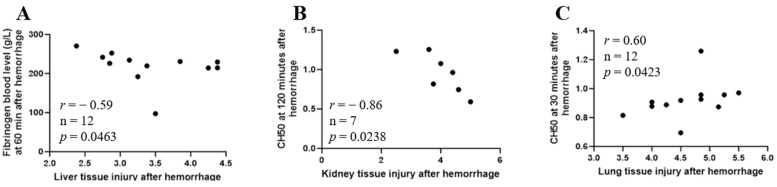
Correlation between circulating fibrinogen/CH50 and end organ injury after hemorrhage. Correlation analysis between fibrinogen/CH50 and liver i (**A**), kidney (**B**) and lung (**C**) organ injury was performed by using Spearman’s rank correlation. Data are shown as individual values with the correlation coefficient (rs).

**Figure 9 ijms-25-02500-f009:**
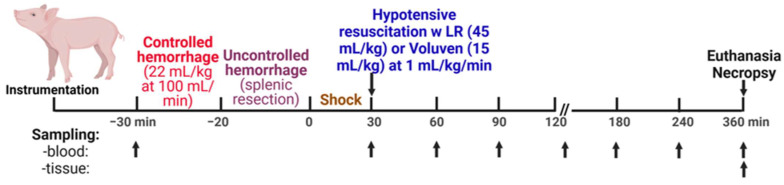
Scheme of the experimental design.

**Table 1 ijms-25-02500-t001:** Hemodynamic response to hemorrhagic shock and fluid resuscitation.

	Group			
	Sham	H	H + LR	H+Voluven^®^
**n**	9	13	12	6
**Body weight (kg)**	39.4 ± 1.1	38.2 ± 0.6	39.1 ± 0.8	40.8 ± 1.7
**Controlled SBV (mL/kg)**	22	22	22	22
**Uncontrolled SBV (mL/kg)** at 30	N/A	11.1 ± 1.0	7.8 ± 0.9	10.1 ± 1.5
60	N/A	12.7 ± 1.1	10.7 ± 2.4	13.9 ± 1.8
90	N/A	12.7 ± 1.1	8.6 ± 1.0	16.1 ± 2.3 ‡
120	N/A	12.5 ± 1.6	8.6 ± 1.0	17.5 ± 2.7 ‡
360 min	N/A	13.7 ± 1.0	11.0 ± 2.3	18.3 ± 2.8 ‡
Fluid resuscitation (mL) at 60	N/A	N/A	733.2 ± 183.1	391.2 ± 94.3
90	N/A	N/A	1057.1 ± 218.0	481.7 ± 106.8
120	N/A	N/A	757.2 ± 237.4	544.1 ± 107.5
240	N/A	N/A	1385.8 ± 241.0	636.8 ± 90.7
360 min	N/A	N/A	1757.3 ± 75.3	785.4 ± 16.4
**PP (mmHg)**				
Baseline	26.7 ± 2.1	28.7 ± 3.2	29.1 ± 2.8	28.8 ± 1.8
At 30	28.5 ± 1.7	20.1 ± 2.9	17.7 ± 2.7 *	19.7 ± 4.0 *
60	29.1 ± 1.3	21.5 ± 3.1	22.0 ± 2.6	27.8 ± 3.1
90	27.2 ± 1.5	14.3 ± 1.5	22.0 ± 3.3	28.0 ± 2.6 †
120 min	26.0 ± 1.5	16.3 ± 3.8	21.3 ± 3	24.6 ± 3.9
**MAP (mmHg)**				
Baseline	63.4 ± 2.5	62.6 ± 3.2	63.4 ± 3.1	60.8 ± 1.6
At 30	65.1 ± 1.3	38.7 ± 2.3 *	39.5 ± 3.8 *	32.2 ± 4.1 *
60	63.3 ± 1.4	37.9 ± 2.2 *	44.6 ± 2.8 *	44.3 ± 1.5 *
90	62.4 ± 1.9	29.6 ± 1.9 *	44.1 ± 1.9 *†	43.0 ± 1.3 *
120 min	63.4 ± 1.8	26.9 ± 2.2 *	43.0 ± 1.6 *†	38.1 ± 3.7 *
**Shock index (bpm/mmHg)**				
Baseline	1.3 ± 0.1	1.4 ± 0.2	1.3 ± 0.1	1.3 ± 0.1
At 30	1.4 ± 0.1	4.5 ± 0.3 *	5.2 ± 1.2 *	4.8 ± 0.7 *
60	1.5 ± 0.1	4.5 ± 0.2 *	3.5 ± 0.4 *	3.2 ± 0.1 *†
90	1.5 ± 0.1	5.4 ± 0.5 *	3.2 ± 0.2 *†	3.1 ± 0.1 *†
120 min	1.6 ± 0.1	5.3 ± 0.2 *	3.4 ± 0.2 *†	3.3 ± 0.1 *†

Legend: Data are expressed as mean ± SEM; n, number of samples; H, hemorrhage; LR, lactated Ringer’s solution; N/A, not applicable; PP, pulse pressure; MAP, mean arterial pressure; SBV, shed blood volume. * *p* < 0.05 vs. sham; † *p* < 0.05 vs. H; ‡ *p* < 0.05 vs. H+LR (two-way ANOVA, GraphPad Prism 5.03). There was no significant difference in body weight between the groups (one-way ANOVA, GraphPad Prism 10).

**Table 2 ijms-25-02500-t002:** Metabolic responses to hemorrhagic shock and fluid resuscitation.

	Group			
	Sham	H	H + LR	H+Voluven^®^
**n**	9	14	12	6
**pH:** Baseline	7.44 ± 0.02	7.43 ± 0.01	7.43 ± 0.01	7.45 ± 0.01
At time 30	7.43 ± 0.02	7.40 ± 0.01	7.40 ± 0.02	7.42 ± 0.03
60	7.44 ± 0.02	7.38 ± 0.01	7.38 ± 0.01	7.36 ± 0.02
90	7.45 ± 0.02	7.38 ± 0.03	7.39 ± 0.01	7.39 ± 0.03
120 min	7.45 ± 0.02	7.40 ± 0.03	7.41 ± 0.02	7.45 ± 0.07
**Base excess** (mmol/L): Baseline	5.3 ± 0.8	6.2 ± 0.6	5.2 ± 1.0	6.7 ± 0.8
At time 30	6.3 ± 0.9	2.5 ± 0.6 *	2.8 ± 0.7	0.4 ± 1 *
60	5.9 ± 0.8	−0.7 ± 1.0 *	1.8 ± 1.0 *	1.6 ± 0.9 *
90	6.1 ± 1.0	−3.4 ± 2.2 *	3.5 ± 1.0 †	3.2 ± 1.2 †
120 min	7.1 ± 0.9	−3.7 ± 1.7 *	4.0 ± 1.1 †	4.0 ± 1.4 †
**Lactate** (mmol/L): Baseline	2.2 ± 0.2	2.0 ± 0.1	2.2 ± 0.2	2.0 ± 0.2
At time 30	1.9 ± 0.1	4.1 ± 0.2 *	3.3 ± 0.2	5.3 ± 0.6 *
60	1.7 ± 0.1	6.6 ± 0.6 *	5.1 ± 0.8 *	5.5 ± 0.4 *
90	1.7 ± 0.2	8.9 ± 1.9 *	4.3 ± 0.7 *†	4.9 ± 0.4 *†
120 min	1.5 ± 0.2	9.3 ± 1.5 *	3.8 ± 0.6 *†	5.2 ± 0.8 *†
**SvO_2_** (%): Baseline	79.0 ± 3.3	73.4 ± 2.4	76.0 ± 1.2	79.4 ± 2.2
At time 30	79.4 ± 2.2	54.3 ± 6.5 *	52.2 ± 6.5 *	66.5 ± 6.4
60	78.2 ± 2.5	62.1 ± 6.5	59.1 ± 3.7 *	71.7 ± 4.2
90	74.4 ± 1.4	71.3 ± 7.4	54.2 ± 5.6 *	71.8 ± 4.3
120 min	77.1 ± 2.6	49.4 ± 10.4 *	56.5 ± 4.8 *	67.3 ± 6.1
**Hemoglobin** (g/dL): Baseline	8.3 ± 0.3	8.4 ± 0.17	8.6 ± 0.2	8.1 ± 0.2
At time 30	8.1 ± 0.4	8.31 ± 0.23	8.6 ± 0.4	7.8 ± 0.4
60	8 ± 0.4	8.25 ± 0.27	6.9 ± 0.6 *	4.5 ± 0.4 *
90	7.9 ± 0.3	6.6 ± 1.0	6.3 ± 0.6	4.6 ± 0.5 ‡†
120 min	8.3 ± 0.3	7.1 ± 0.7	6.7 ± 0.37 †	4.4 ± 0.4 †*
**Hct** (%): Baseline	24.5 ± 1.06	24.7 ± 0.5	25.4 ± 0.6	23.8 ± 0.8
At time 30	25.2 ± 1.2	24.5 ± 0.7	25.3 ± 1.2	23.0 ± 1.2
60	24.2 ± 1.4	23.2 ± 1.3	19.4 ± 1.8	13.2 ± 1.0 *†‡
90	23.5 ± 1.2	19.4 ± 2.8	18.4 ± 1.6	12.8 ± 1.4 *†‡
120 min	25.2 ± 1.2	21.0 ± 2.1	19.7 ± 1.1 *	12.4 ± 1.1 *†‡
**Potassium** (mmol/L): Baseline	4.1 ± 0.0	4.0 ± 0.1	3.9 ± 0.1	3.9 ± 0.1
At time 30	4.2 ± 0.1	4.7 ± 0.1	4.6 ± 0.3	5.1 ± 0.3 *
60	4.2 ± 0.1	5.1 ± 0.2 *	4.4 ± 0.3†	3.8 ± 0.1 †
90	4.4 ± 0.1	5.2 ± 0.4	4.4 ± 0.1	4.3 ± 0.1
120 min	4.5 ± 0.1	6.2 ± 0.7 *	4.6 ± 0.1 †	4.8 ± 0.1 †

Legend: Data are expressed as mean ± SEM; n, number of samples; H, hemorrhage; LR, lactated Ringer’s solution; SvO_2_, mixed venous oxygen saturation. * = *p* < 0.05 vs. sham; † = *p* < 0.05 vs. H; ‡ = *p* < 0.05 vs. H+LR (two-way ANOVA, GraphPad Prism 10).

## Data Availability

All data generated or analyzed during the current study are included in this published manuscript. The data presented in this manuscript are available on request from the corresponding author.

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
