# Peer review of "Impact of Immunopathy and Coagulopathy on Multi-Organ Failure and Mortality in a Lethal Porcine Model of Controlled and Uncontrolled Hemorrhage"

_ijms, 2024, doi:10.3390/ijms25052500_

Round 1

Reviewer 1 Report

Comments and Suggestions for Authors

The authors have measured hemodynamic, metabolic, immunoinflammatory and organ function markers in a pig model of controlled/uncontrolled hemorrhagic shock with LR/HES resuscitation. The following comments/queries require clarification and amendments to the manuscript.

Title: The title is not reflective of the key findings of this study. No statistical analysis was conducted looking at the association between immunopathy and coagulopathy with MOF and mortality and the study seems to focus more on the effect of fluid resuscitation (LR vs HES).

Abstract: The methods need to include the monitoring period. Also the methods don't reflect the polytrauma model described in the Materials and Methods of the manuscript with soft tissue injury/femur fracture. The conclusion is not supported by the data as the results don't indicate a significant coagulopathy in the HS alone group (only resulting from fluid resuscitation). To make this statement of an early coagulopathy following hemorrhagic shock would require within group analysis of the HS alone group for these parameters which was not completed. Furthermore, there was no specific statistical analysis to assess associations between immunopathy and coagulopathy with MOF and death. DCR is not mentioned in the Abstract but is a feature throughout the manuscript.

Introduction: Paragraph 2 jumps between fluid resuscitation then transfusion then DCR which is not sufficiently explained for a general audience.  References are required to support the statements in paragraph 3. Define Bb biomarker.

Materials nad Methods: Specify whether pigs were SPF. Clarify whether baseline screening was done prior to surgical instrumentation (e.g., via peripheral ear catheter). The effect of surgical trauma on these parameters should be considered a limitation if baseline measures were taken after all surgical procedures. The size and approach of the laparotomy should be specified and whether it was closed immediately after splenic resection as this has implications for the progression of internal hemorrhage. The methods refers to some animals having soft tissue trauma and femur fracture before splenic injury. The results are subsequently not divided by animals that had splenic injury alone or those with polytrauma/splenic injury. The time from injury to fluid resuscitation needs to be included in the text of the methods. There appears to be some repeats in the discussion about euthanasia under Biosampling. Specify specific tissue harvest, e.g., right/left lung, kidney. Liver is not  included in the tissue harvest but is presented in the results. Specify the number of sections per sample were analysed for histological examination. Specify whether cytokines were measured in duplicate as per literature standard and adjusted for total protein (and method used to determine total protein). IL-10 is mentioned in results but not included in methods. How was sample size determined and why is there a discrepancy in group sizes? Statistical analysis does not include any evaluation of data normality to determine appropriate statistical test to use. Statistical analysis does not reference how histological data was analysed.

Results: Survival needs to be clearer from the start and specific n values for each parameter and timepoint need to be included in all tables and figures. Table and figure legends should include explanations for lower n values than expected (e.g., Table 1 H group n=13 rather than 14 and H + LR group n=12 rather than 13). Why are no hemodynamic parameters reported for 240 min and 260 min timepoints? Table 2 - pH should be presented to at least 2 decimal places for clinical relevance. The total volume of fluid resuscitation for LR and HES groups should be presented (so readers can compare to total blood loss). Fig 1 - based on a visual examination of the graphs coagulopathy was associated with fluid resuscitation and not hemorrhage alone. In order to report that HS led to coagulopathy, a repeated measures ANOVA or equivalent within groups analysis should be conducted looking at changes in coagulation parameters over time (rather than just overall between groups analysis). Similarly, the increased systemic complement function is linked to HES and LR resuscitation, rather than hemorrhage alone. Fig 6 - provide magnification used.

Discussion: The discussion includes more background/literature review than critical analysis of the results of the current study. Critical analysis is lacking. The discussion suggests that this study incorporated DCR however this is not clear in the methods. The discussion of TXA does not reflect published literature that has reported that TXA can worsen immunoinflammatory responses. The paragraph on histological splenic examination is very poorly linked to the previous paragraph. It also reports an observation period of 24 h which is inconsistent with the methods and results. Histological splenic examination is also not reported in the methods. The statement that fixed-volume models of HS more closely simulation hemorrhage seen in accident victims or combat casualties is inconsistent with literature and the introduction. A major justification for the study is that controlled hemorrhage models are not clinically relevant however this study incorporates controlled hemorrhage within its model. This needs to be discussed as a limitation. There are multiple other limitations of the study that require discussion, including the use of only female pigs, the known differences between pig and human coagulopathy, maintenance of core temperature throughout with a homeothermic blanket (this does not reflect the clinical scenario and temperature has an impact on coagulopathy), the use of laparotomy which removes the natural tamponade provided by an intact abdominal wall, very early mortality not showing organ injury on histological analysis.

Reviewer 2 Report

Comments and Suggestions for Authors

With great interest, I read a work from Simovic et al. on the impact of immunopathy and coagulopathy on multi-organ failure and mortality in a lethal porcine model of uncontrolled hemorrhagic shock. Authors should be congratulated on a well-performed work.

Please address in the abstract the difference in outcome in regard to use of Voluven and LR

The first paragraph on page 2 is a discussion paragraph and not an introduction. This should be adapted.

Creatinine (a biomarker of kidney function)” please remove the text in ().

In the section “2.10. Effect of Hemorrhage and Fluid Resuscitation on Survival” and the whole text should be adapted for this, no phrases like “As you can see in” should be used.

The first paragraph of the discussion is a repetition of the introduction. This should not happen. The whole discussion should be adapted for similar cases.

A recent study with a similar aim should be cited (https://pubmed.ncbi.nlm.nih.gov/36832127/)

Comments on the Quality of English Language

English language and flow should be adapted. I would recommend a native speaker who was not involved into work to read it and recommend adaptations. The work is lacking standard scientific english 

Reviewer 3 Report

Comments and Suggestions for Authors

The study examined both controlled and uncontrolled models of hemorrhagic shock in a swine model. It compared the efficacy and safety of Ringer Lactate (RL) and Voluven regarding inflammatory response and organ function. The comprehensive analysis covered systemic inflammatory response, myocardial inflammatory response, as well as inflammatory responses in the jejunum, lungs, and kidneys.

Additionally, the authors investigated the fluid type's impact on survival, noting a clear benefit associated with the application of Ringer Lactate (RL). This study holds promise for future advancements in human therapy. However, I have some concerns:

1. The title is misleading for the reader: The title suggests an assessment of the impact of coagulopathy and immunopathy on MOF. However, the study primarily analyzes the impact of two different fluids in the fluid resuscitation of two types of shock. While the study's results are compelling, the title lacks consistency with the actual study content. The most significant finding is that RL seems more effective than Voluven in the aspects presented. The authors did not investigate direct correlations between immunopathy and coagulopathy on MOF. I think, that the title having information about fluid resuscitation strategies would be more appropriate.

2. 'Immunopathy' appears somewhat vague. I would expect this term to refer to immunosuppressed or immunologically knocked-out animals, or specific lymphocyte function tests. A better option might involve using inflammatory markers or cytokine responses.

3. The study is well-designed, employing two controls: hemorrhage and sham.

If the intention is to assess what the title suggests, it would be essential to correlate MOF markers with inflammatory markers and clotting parameters. Such results appear to be lacking.

4. Abbreviations: The 'DAPI' abbreviation is introduced for the first time in Site 17, yet it's used in Figure 4. It would be beneficial to define this abbreviation there.

5. The study's aim should be modified, or additional analyses are required to establish correlations between inflammatory and thrombotic parameters and markers of MOF.

6. Similarly, the conclusion stating, 'Controlled and uncontrolled hemorrhage-triggered immunopathy and coagulopathy were associated with increased MOF and mortality,' needs refinement to align with the study's actual findings."

Round 2

Reviewer 1 Report

Comments and Suggestions for Authors

Thank you to authors for addressing all of the comments and making appropriate change to the manuscript.